# Evidence-Based Implementation of the Family-Centered Model and the Use of Tele-Intervention in Early Childhood Services: A Systematic Review

**DOI:** 10.3390/healthcare12010112

**Published:** 2024-01-03

**Authors:** Estibaliz Jimenez-Arberas, Yara Casais-Suarez, Alba Fernandez-Mendez, Sara Menendez-Espina, Sergio Rodriguez-Menendez, Jose Antonio Llosa, Jose Antonio Prieto-Saborit

**Affiliations:** University Clinic INYPEMA, Faculty Padre Osso, University of Oviedo, 33008 Oviedo, Spain; estibaliz@facultadpadreosso.es (E.J.-A.); yaracasaissuarez.to@gmail.com (Y.C.-S.); albafmendez@outlook.es (A.F.-M.); saramenendez@facultadpadreosso.es (S.M.-E.); s.neuronast@gmail.com (S.R.-M.); josea@facultadpadreosso.es (J.A.P.-S.)

**Keywords:** early intervention, telepractice, family-centered care, systematic review, childhood

## Abstract

Background: The purpose of this review is to explore the evidence and efficacy of two trends in early childhood intervention services: the family-centered model and the use of tele-intervention. Methods: A systematic review was carried out following the PRISMA methodology and using three databases: Web of Science, PubMed and Scopus. The studies included were those aimed at children from 0 to 6 years of age, focused on early intervention, and which alluded to the family-centered model and/or tele-intervention. Results: a total of 33 studies were included. Five main themes were identified: (1) The participation of children and family is facilitated and improved by the family-centered model of care; (2) the feeling of competence, self-efficacy, satisfaction and empowerment in professionals and families have a positive impact on quality of life; (3) the use of tele-intervention as a tool for prevention and intervention; (4) preparation for telepractice can improve the development of commitment; (5) tele-intervention as a possible solution to contextual barriers. Conclusions: Tele-intervention in pediatrics is presented as a tool inherent to the family-centered model since its implementation involves several common strategies. Future lines of research should explore the use of this tool as a possible solution to contextual barriers.

## 1. Introduction

Infant maturational development is the name given to a series of physiological processes that enable the maturation, organization and function of the different apparatuses and systems that together make up the human organism [1,2,3]. Children, through their performance and participation in different activities, experience and interact with the environment by developing the necessary and expected motor, cognitive and sensory-perceptual skills, in order to face increasingly complex challenges, according to their culture, society and age [2]. This suggests that there is a relationship between the number and type of experiences gained and the need for support in different aspects of life. The scientific literature provides descriptive information about children with neurodevelopmental disorders and the appearance of difficulties in the acquisition and deployment of specific intellectual, motor or social functions. This will negatively impact the participation in, and performance of, daily life activities [4,5,6], resulting in their restriction and limitation, respectively [7].

Early intervention means identifying and providing effective early support to children who are at risk of poor outcomes [8]. In these children, it has been shown to improve school performance [9,10,11], as well as skill development and acquisition [12]. Therefore, it is essential to identify and address those who present some alteration in their development or who are at biological, social or environmental risk. This care is offered between the ages of 0 and 6 years in order to improve these capacities and skills, as well as to prevent or reduce the impact of possible difficulties that may arise throughout their lives [2,13,14]. There is also evidence that children who are developmentally delayed or have a diagnosis that affects their development, and who live in small rural municipalities, face unique challenges due to limited access to specialized resources and services [15]. Therefore, there is a pressing need to analyze and research new trends in Early Intervention for these environments and this type of population [12,14,16,17].

Within contemporary trends in Early Intervention, one of the tendencies is based on the involvement of families in supporting the early development of the child, which is essential to overcome the negative consequences associated with disability or developmental delays [18]. With a purely ecological and transactional model, working with the family must be one of the pillars in any system working with early childhood [19,20,21]. This trend is known as family-centered care, the key principles of which are empowerment in decision-making [22]; increased levels of competence and involvement of the family, and the strengthening of the family’s bonds with the practitioners involved in the process [20,23]. The practical application of these principles is positively related to the development of children with disabilities [24]. Therefore, families are considered central elements during the intervention process, who must be guided and trained by the team of professionals who intervene from the family-centered care model.

The introduction of this model into childcare practices is incorporated as the possibility for families to use their specific circumstances to optimize developmental opportunities and thereby increase the child’s quality of life [22]. It differs from the previously established expert model, in which the family is relegated to a secondary role [19]. In fact, in the more classic scientific literature, it is already shown that families play a fundamental role in the field of early intervention because of the time that the family shares with the child [25]. For this reason, they are considered to be fundamental in identifying and responding to the communicative signals of their children [26]. The fundamental principle underpinning intervention based on the family-centered model is to build on the needs, concerns, wishes and expectations of the caregivers themselves, working in natural settings and contexts, not just in a specialized intervention room. The aim is to facilitate real experience-based learning and provide the opportunity to generalize skills into the routines that make up the day-to-day life of families [20,23,27]. Components such as motivation are of vital importance, as an indication of commitment or a feeling of capability [28,29].

Traditionally, family-centered interventions have been carried out face to face, where the practitioner and the family are in the same room [20]. Despite having suggested a practice based on the family-centered model of care, certain factors may hinder access to specialized services. The variables mentioned in the scientific literature and considered to be the most limiting are based on the shortage of specialized professionals, problems accessing transport and the overload of services with long waiting lists [15,18,30]. Such variables are exacerbated in more remote settings. Again, the conditions of rural–urban disparity become notable and bring to light accessibility barriers that directly and negatively affect the response to the care needs required by families. Sanz Tolosana et al. [31] conclude that it is necessary to consider inclusive policies that truly take into consideration the different particularities of the rural population and their territories. Recent studies refer to the use of alternative tools or methods to conventional actions that take advantage of the development of technological resources, something which is currently in constant growth. In the healthcare context, this is known as tele-intervention.

In response to these challenges, and with the advent of the global health emergency caused by COVID-19, the need to find alternative solutions to the existing ones has become apparent in order to ensure continuity and proper development of care services. In this sense, the use of Information and Communication Technologies (ICT) has provided a feasible option to guarantee access to different intervention services in the sphere of healthcare [32,33]. The application of this technology in the provision of healthcare services is known as tele-intervention, including interactive services for both remote consultation and remote diagnosis [34]. Superficially, tele-intervention does not differ much from expert model’s methodology, with the interaction between the child, family, practitioner and support staff being the common thread of the intervention [18]. In the study by Olsen et al. [35], they compared the interactions between different participants during both telematic and face-to-face sessions. On the one hand, family coaching was conducted simultaneously with the session with the child, in order to discuss and select strategies to facilitate the child’s development and acquisition of skills. On the other hand, a direct interaction with the family and the child took place. This second procedure allowed for continuity in the sessions while the family received information and comments in situ from the practitioner, making it possible to generalize learning to the real environment. Statistically significant differences were found between the intervention-defining behaviors in the different groups.

Given the imminent paradigm shift in early childhood care in recent years, compounded by the global pandemic of 2020, there is a need for continued research in this area.

### This Study

This study aims to explore and synthesize the existing scientific evidence surrounding two types of interventions implemented in the field of Early Childhood Intervention: one based on the Family-Centered Model and the other on tele-intervention, both targeting pediatric populations with developmental issues or at risk of experiencing them. It focuses on exploring studies involving individuals aged between 0 and 6 years old, aiming to enhance levels of independence and functional autonomy. Additionally, it emphasizes the search of disparities in the form of intervention between rural and urban environments. The two types of intervention have been chosen for the same study due to their complementary nature of bringing Early Intervention closer to families. In addition, they are having a growing interest at a professional and a research level. Therefore, the question arises as to whether the models that bring Early Intervention closer to the family show scientific evidence and what benefits they bring to the intervention.

The main objective is to understand the scientific evidence regarding the efficacy of tele-intervention and the Family-Centered Model in Early Childhood Intervention. To achieve this, two specific objectives are presented: explore the effectiveness evidenced in the use of the Family-Centered Model in early childhood care practices and examine the effectiveness and effects of interventions that use tele-intervention.

## 2. Materials and Methods

### 2.1. Study Design

A systematic review was conducted to explore the degree of effectiveness of early interventions based on the family-centered model and tele-intervention. To this end, we worked on the basis of the updated Preferred Reporting Items for Systematic Review and Meta-analyses (PRISMA) 2020 guidelines [36]. Use is made of the International Classification of Functioning, Disability and Health (ICF) [7,37] nomenclature to express the results obtained in the search. In an initial phase, and with the aim of offering a contemporary view of the topic, the last complete decade was taken as a reference for the search for studies, that is, from 2012 to 2023.

### 2.2. Search Strategy

A combined strategy of electronic searches was carried out in order to identify target studies for the research. These studies had to have been published between 2012 and the date of the search in the following databases: Web of Science, PubMed and Scopus. The search terms were differentiated into two distinct phases under the same central axis that coincides with the main theme of the study: the Early Childhood Care resource. The first search phase corresponded to early intervention and the family-centered intervention model. The second search phase consisted of linking the concept of early intervention with tele-intervention. In both phases, the Boolean operators AND, NOT and OR are combined to identify potential articles that integrate the aforementioned topics in the same study. The searches carried out in each of the databases are described in detail in Appendix A. Subsequently, the different titles and abstracts were reviewed to determine whether the studies were related to the main objective of this research and the inclusion and exclusion criteria described below.

### 2.3. Selection Criteria

The catalogue of services and benefits varies according to the different territorial contexts. This also affects the definition of the main focus of this research. For the purposes of this systematic review and possible future lines of research, early intervention will be understood as the set of interventions aimed at the child population aged 0–6 years, the family and the environment, responding as early as possible to the temporary or permanent needs of children with developmental disorders or at risk of suffering from them.

All studies included in this review are subject to meeting the following criteria: (1) the interventions carried out are framed within the framework of early childhood care; (2) variables related to the effectiveness of interventions based on the family-centered model and/or tele-intervention were investigated; (3) the different perspectives of one or more of the following possible groups involved were explored: family, infants, professionals; (4) a sample given by any number of infant individuals was described; regardless of cause, gender, ethnic group or geographical location with an age between 0 and 6 years; (5) the study is available in English and/or Spanish; (6) studies that dealt only with purely biomedical interventions that did not consider psychosocial aspects, such as the different perspectives of the groups that could be involved in the process, were excluded; (7) studies that were developed in contexts of high specificity (e.g., Neonatal Units) were excluded; (8) documents whose development and ultimate aim consisted of the creation of protocols or good practice manuals; and (9) conference proceedings, as they were not certain to have been peer-reviewed, were excluded.

### 2.4. Selection of Studies

The search results were imported into the Zotero bibliographic manager [38] version 6.0.26 for the removal of duplicates and then screened by title/abstract. The full texts of all the studies that passed the first selection stage were reviewed to assess their eligibility according to the inclusion and exclusion criteria. In parallel, in order to extract the relevant information for the study, a content analysis (i.e., an in-depth reading of the studies) was performed to obtain a list of the most decisive and defining data.

### 2.5. Quality Assessment and Risk of Bias

A classification of the levels of evidence and grades of recommendation of the selected articles based on the Scottish Intercollegiate Guidelines Network (SIGN) has been included to assess the quality of the studies [39]. This guide classifies the level of evidence according to the type of study design in an interval from 1 to 4, with the symbols “++”, “+” and “−” in scores 1 and 2, in order to further detail this level of evidence. In addition, based on this first classification, this guide allows the establishment of grades of recommendation in the interval A–D, with the study classified as “A” resulting from a direct recommendation.

This tool makes it possible to reduce different types of bias, as its authors indicate, having been used in previous systematic reviews [40]. Likewise, to specifically avoid the risk of individual bias, two members of the team separately made the final choice of studies to include. Those in which there was no consensus were discussed with a third reviewer. Subsequently, parallel processes were also carried out to decide the relevant variables to record from the articles, with subsequent discussion of cases where there was no consensus. For all these processes, Microsoft Excel version 16.74 software was used.

## 3. Results

The results of the search in the different databases and the next steps applied can be seen in the flowchart represented in Figure 1. The original search yielded a total of 221 potential studies. A total of 29 duplicate publications were then eliminated. Of the remaining 192 potential articles, a first screening was done by reading the title and abstract, discarding a total of 137 publications; a total of 55 articles were proposed for full reading. Finally, 33 studies were included in the review, having met the inclusion criteria.

Of the publications finally selected for analysis, 15 (45.45%) dealt with the Family-Centered Care Model; 12 (36.36%) with tele-intervention and 6 (18.2%) with both. Of the total number of studies, 85% were quantitative, 6% were qualitative and 9% used a mixed methodology. These studies were conducted from different geographical locations worldwide, corresponding to the following percentages and represented in Figure 2: United States (33%), Spain (18%), Australia (9%), Netherlands (6%), China (6%), India (3%), Italy (3%), South Africa (3%), France (3%), Taiwan (3%), Republic of Korea (3%), United Kingdom (3%), Ghana (3%) and Norway (3%). Furthermore, different disciplines were involved in the research, on the one hand: physical therapy, occupational therapy, developmental therapy, speech therapy, physiotherapy, psychology, social work, pedagogy and neuropediatrics, and on the other, a variety of participants, with three main groups: family members or primary caregivers (63%), infants (24%) and clinicians (18%), with the first target group being the most predominant. Some of the studies included several of these samples simultaneously. A variety of diagnoses were also described: autism spectrum disorder (9%), visual impairment (1%), hearing impairment (18%), deafblindness (3%), motor impairment (3%), developmental delay (15%), as well as other variables or factors related to the different interactions between the practitioner, the family and/or the child.

The record of the extraction of these data is summarized in Table 1. Likewise, the data concerning the degree of evidence of these studies are reflected in Table 2. 

### 3.1. Data Extraction and Synthesis

As a result of the thematic synthesis process, five main themes were identified that are in line with the objectives of the study: (1) the participation of children and family is facilitated and improved by the family-centered model of care; (2) the feeling of competence, self-efficacy, satisfaction and empowerment in professionals and families have a positive impact on quality of life; (3) the use of tele-intervention as a tool for prevention and intervention; (4) preparation for tele-intervention can improve the development of commitment; and (5) tele-intervention as a possible solution to contextual barriers (Figure 3).

This thematic synthesis has been carried out through the subtraction of the different study variables and the results provided by the authors, finally pooling the different themes mentioned above. The reason why the results are presented in this way is to illustrate the information presented in Table 1 in order to facilitate reading.

#### 3.1.1. The Participation of Children and Family Is Facilitated and Improved by the Family-Centered Model of Care

The family-centered model of care was described as a facilitator in establishing a link with the clinical environment in the early intervention setting. This approach allowed both children and family to feel safe and motivated throughout the process [21,41,42,44,59,64,65].

Thanks to the principles that guide this model [22,25], the establishment and subsequent maintenance over time of the bonds of the therapeutic relationship is made possible. A large number of the families described the implementation of this model as positive, showing higher levels of satisfaction, confidence, competence and empowerment with the service [42,44,64].

The study by Dick et al. [56] showed a comparison between the early intervention service and the primary care service. In the former, the context invited an understanding of the family as an active agent and, therefore, their inherent inclusion in the intervention plan from the very beginning. This generated an approach to the domains of habituation and participation which, in turn, translated into satisfaction, self-efficacy and positive perception of the service on the part of the families. The second, on the contrary, a more biomedical context, entailed placing the child as the sole focus of intervention, differing the practitioner–caregiver relationship from the previous context.

#### 3.1.2. Feelings of Competence, Self-Efficacy, Satisfaction and Empowerment in Practitioners and Families Have a Positive Impact on Quality of Life

Practices based on the family-centered model ensure family commitment and are related to positive therapeutic outcomes [41,42,43,44,48,53,59,61,63,64,65]. As shown in the previous section, variables such as competence, self-efficacy, satisfaction, and empowerment are of great importance since, to some extent, the intervention will depend on them. Taking into account aspects such as the family’s beliefs, values and needs, as well as offering individualized support and guidance [43], are practices that favor the appearance of higher levels of the aforementioned variables, which is why their promotion is necessary and in turn generate a good therapeutic relationship [20,23].

Although there is a desire to generalize the family-centered model in current practices [22], different barriers are described that are independent of the practitioners, such as lack of time, resources, training, or aspects related to the organization and structure of the system [21,60].

On the other hand, the results obtained in the study by Cheung et al. [54] point to the need for practitioners to strengthen their knowledge and skills to provide a service based on the family-centered model. The ultimate goal is to empower caregivers by creating learning opportunities for this learning to be generalized and applied to the natural environment [72]. The findings suggest that an intervention based on these principles promotes and helps to achieve the child’s goals, as well as higher levels of caregiver satisfaction and internal motivation [63,65].

#### 3.1.3. Use of Tele-Intervention as a Tool for Prevention and Intervention

In the scientific literature, the use of tele-intervention was described in the healthcare context as an alternative tool to conventional interventions. The use of technology in this field is becoming increasingly common in assessment and intervention processes. This can be seen in the progressive increase in the number of publications that include this term in the Web of Science (Figure 4). The casuistry, the circumstances and the approach from which tele-intervention is used entails the application of techniques or forms of action that are inherent to the family-centered model: active family participation, attention to the needs and priorities demanded, joint search for solutions [42,51,63].

The methodologies used in the community healthcare field are backed by research that ensures evidence-based clinical practice with empirical results. The implementation of new techniques or intervention models not only requires theoretical validation to justify and provide quality but also social validation.

The recent study by Martínez-Rico et al. [42] conducted in a Spanish population analyzes factors of feasibility, usefulness or possible future interventions of tele-intervention in early intervention services. It also takes into account usability, effectiveness, competence and trust, factors that are relevant to what the authors call social validity. Other studies such as those by McCarthy et al., Popova et al., García-Ventura et al., du Plessis et al., Rose et al. and Vilaseca et al. [20,21,52,58,60,66] assess factors that have to do with the technical staff who carry out the intervention, seeking their professional validation.

The global health emergency caused by COVID-19 compromised the vast majority of methodologies and techniques that were carried out in a traditional, face-to-face manner, leading to a change in their modality [73,74]. In this context, protocols and guidelines were developed to facilitate this shift towards tele-intervention. The use of tele-intervention became very important and was done with different objectives. On the one hand, a preventive vision was taken, where the objective was to provide quality information and raise awareness among the population, as well as strategies for maintaining a healthy lifestyle. On the other, from a more interventional point of view, new forms of practitioner–user communication were facilitated, and the diagnostic and treatment process was adapted without the need for travel. In other words, the aim was to act in the natural environment [75].

In the systematic review process, 2 studies were detected where the terms “COVID-19” and “Telehealth” are found in the title, and both describe the use of tele-intervention technology as an intervention tool with positive results in children [75]. Miller et al. [50] report closer follow-up of neonates after discharge from the intensive care unit, which led to increased detection of cases requiring additional referral to another service. In the paper by Kronberg et al. [63], technicians worked with a coaching-based intervention with families for 9 months. The findings suggested that the use of tele-intervention in that procedure was effective in achieving goals. A subsequent study published by Qu et al. [45] in a population with Autism Spectrum Disorder (ASD) used the same intervention method and obtained positive results in terms of family perception variables.

Subsequent research continues this line of prevention and intervention in determining aspects of the health of infants aged 0 to 6 years, showing rates of effectiveness and acceptability, in which the figure of the family and its context are highlighted in the process [42,45,51,53,54,57].

#### 3.1.4. Communication during Tele-Intervention May Be Limited by Logistical Barriers

A previous training and interaction stage prior to the use of the tele-intervention methodology could improve the development of commitment and communication between families and practitioners. In this sense, there are two main groups to take into account: the family and the technical team.

The study by Yang et al. [62] shows reluctance about the use of telehealth in the early intervention service. Several family members reported feelings of discomfort and inability to conduct sessions remotely or skepticism that professionals could solve situations remotely. One possible explanation put forward was that the family was seeking a child-centered approach to intervention where the use of face-to-face materials with the therapist was viewed as more valuable. Other barriers to its implementation also come into play, which are consistent with the studies by Rose et al., Blaiser et al., and Li et al. [66,69,70], and are limiting for the proper functioning of the sessions. These refer to access to a line with unlimited data, broadband internet connection and electronic devices, as well as the ability to manage them.

Another barrier they highlight and which is directly related to family perceptions is adherence. Research by Li et al. [69], which made use of SMS text messages as a tool for collecting data about health programs in rural China, showed a low participation rate justified by participants as forgetfulness. In these situations, Staiano et al. [57] propose the use of automatic notifications to reinforce the sample’s use of the data collection platform or system. This method of notifications was also implemented by the team of Denis et al. [53] in order to create a more efficient case detection and referral system.

As for the professional branch, the increasingly common use of tele-intervention generates controversy on certain occasions. It is conceivable that professionals should have prior specialized training in order to be able to carry out interventions correctly. For example, McCarthy et al. [20] provided continued training in tele-intervention to professionals, adapting it to their needs and following their own protocol created years ago [76], without finding significant differences with respect to face-to-face work. In this case, it was an intervention based on the family-centered model. For their part, Popova et al. [21] advocate for future research to provide training opportunities, specifically to foster communication with families.

#### 3.1.5. Tele-Intervention as a Possible Solution to Contextual Barriers

The use of tele-intervention has demonstrated benefits not only in purely interventional values, but there is also documentation that reflects a more logistical nature. Its application has been implemented in more remote settings or those with fewer resources as a possible solution to continuous transfers to referral health centers, usually located in urban centers. In this systematic review, 3 studies were conducted in rural or developing settings: 2 in different parts of Africa; Ghana and South Africa and 1 in China.

Both the study by du Plessis et al. and Ameyaw et al. [52,67] focused on the detection and diagnosis of deafness or hearing loss. The first study involved the establishment of an interregional network with the ultimate aim of conducting a comparative analysis between two hearing screening techniques, carrying it out remotely and face to face. The results obtained showed high rates of efficiency in terms of the time factor. Thus, families did not have to make such a long journey, which resulted in savings in time and money. The second study, focused on health professionals, discusses the use of tele-intervention as a support for the knowledge and perception of children’s hearing health, which will lead to a scalable and low-cost intervention in the prevention of childhood hearing loss or deafness. In the study by Li et al. [69], although adherence to the monitoring program was not adequate, resulting in lower response levels than the face-to-face method, the cost was lower.

## 4. Discussion

This systematic review aimed to describe the nature of the family-centered model and tele-intervention methodology in early childhood intervention, specifically in the 0–6 age group. At this stage, there are important developmental changes [77,78]. The maturational development of the child consists of a series of pre-established developmental patterns that are a consequence of the relationship between intrinsic factors, linked to genetics, and extrinsic factors, linked to the context and environment surrounding the child [1,3,37,79]. This process can be explained through the comprehensive model of functioning, disability and health [7,37], which is defined by three fundamental axes: bodily functions and structures, activity and participation, depending on the health condition. When there is any kind of difficulty in any of the three axes, this is known as impairment, limitation or restriction, respectively. In the event that the triad of factors is altered in its totality, this is when we will speak of disability [7,37].

If this situation occurs, the child population has care services made up of a team of experts in the pediatric area who will be in charge of providing care to those children who are at risk or who suffer from some type of affectation that restricts their capacity for participation and occupational performance, negatively affecting their quality of life [80]. The intervention of these teams can be based on different models that define the clinical practice. In the case of the family-centered model of care, its principles [22,25] facilitate the establishment and maintenance of the therapeutic bond, which enables both families and children attending the service to have positive feelings throughout the intervention process [21,41,42,44,55,59,64,65]. The inclusion of the family as an active agent in the process is the fundamental principle that relates to the intentional relationship model where the interpersonal dynamics or the interaction between the practitioner–family–child triad has the power to enable or inhibit that active participation during the process [21,55]. This is different from other models of care such as the person-centered model [81] or the system-centered model [82], where interaction with the family is merely relational [47].

The authors agree that the perception of the service is conditioned by the quality of the therapeutic relationship. When the practitioner takes into account the values, beliefs and needs of the family [43], as well as their demands and concerns [42] when agreeing on intervention goals, and provides a safe environment where support and guidance is offered [52,59,70], satisfaction, confidence, the sense of competence and empowerment increase [41,44,45,46,64]. These variables are related to the family quality of life, which is considered the ultimate goal of early intervention [59], being important predictors of service quality and social validity [42]. Likewise, the role of practitioners is of vital importance; the scientific literature points to the need for specialized training to improve skills and the development of interdisciplinary competencies for the proper functioning of early intervention services [21].

In recent years, the scientific literature has undergone a process of reorientation and paradigm shift [82], which has entailed a change in the modality of care. These changes imply a profound transformation at the organization level of professional roles and the functioning of the system [81,83,84]. A key example in 2020 was the introduction of tele-intervention on a massive scale, as a solution to the health emergency in the wake of COVID-19, which has been implemented in the field of Early Intervention using different strategies. In the study by Cason et al. [71], they highlight the incorporation of tele-intervention tools in a variety of care programs in the different states through the development of new education and research policies, and the development of secure applications, which are important for improving the quality of care services.

One of the strategic axes is related to prevention and diagnosis, where the aim is to provide quality information and raise awareness among the population to maintain a healthy lifestyle. Rose et al., Helle et al. and Pearson et al. [49,66,68] focus their research on the promotion of healthy eating habits, where they use tele-intervention methodology to transfer information of this nature. Along the same lines, studies more focused on early detection have implemented this technology to collect information based on developmental milestones or standard values of typical development. Thus, in the case of detecting any kind of deviation, this entails notifying the primary caregiver [50,53,67] and suggesting that they request an appointment with their referral center [50,53], enabling them to address it early.

Another axis is the intervention itself, where tele-intervention has been applied in different population groups and with different intervention techniques. The studies by Kronberg et al., Qu et al. and Staiano et al. [45,57,63] base their approach on the coaching technique, showing a good acceptance by families and effectiveness in terms of achieving goals and milestones. This technique is related to the facilitation of a safe environment in which support and guidance are provided [52,59,70] based on active listening and a tandem between the family and the practitioner. On the other hand, Blaiser et al. [70] and du Plessis et al. [52] focus their studies on the deaf and hard-of-hearing population, obtaining positive results in children’s language skills. Landolfi et al. [51] base their study on professional training, establishing a relationship between the level of specialization and the quality of support in interventions with hearing-impaired children. This study is related to the one by Popova et al. [21], which establishes knowing how to communicate as part of the specialization process in the family-centered model, and to that by Cheung et al. [54], which states that it is necessary for practitioners to strengthen their knowledge and skills to collaborate with families and to educate them during sessions.

Likewise, although there is a preference towards the family-centered model, in practice [60], there are barriers or impediments that make its implementation difficult [60,62,70]. In the case of rural areas or developing countries, the population faces unique challenges [85] where tele-intervention seems to promote solutions in terms of providing opportunities [52,67,69].

## 5. Conclusions

The most current literature available alludes to the family-centered model as a purely ecological and transactional paradigm, where active work with the family core is the fundamental pillar when intervening with a child. The ultimate goal of such intervention is to enhance the quality of family life. On the other hand, tele-intervention offers the possibility of being implemented in all phases of the early childhood care process: prevention, diagnosis, and intervention, regardless of the context in which they are established. This allows for an increase in service quality, especially in rural areas that are distant from urban centers.

### 5.1. Limitations of the Study

It should be noted that there are some limitations within this systematic review. Firstly, the use of only three databases could have affected the findings, indirectly excluding some publications that could have been of interest. However, it was decided to select the most relevant databases in the field of study. Secondly, the scope of the review included only articles published in English and Spanish, which may have excluded studies that could have been analyzed in other languages. In addition, studies were included in a time range from 2012 to 2023 in order to show evidence from the last decade, although some studies may have been excluded if they were published previously. Finally, there is a lack of agreement in the literature for the use of terminology that was key in the search for “tele-intervention”, with a large number of expressions to refer to it, and some studies may have been excluded because they did not include the key word. In addition, a meta-analysis study would complete this research to determine with statistical tools the level of evidence of these two types of interventions. It would also allow comparison with other Early Intervention models or analyze potential moderators of the effect of the interventions.

### 5.2. Implications for the Practice and Future Lines of Research

The findings obtained in this systematic review could support the implementation of the family-centered model in early childhood intervention services, advocating for effective communication between the practitioner and the family, and the creation of a strong therapeutic bond that facilitates the management of the child to achieve goals in the natural environment. Along this line, further exploration of the development, maintenance and implementation of tele-intervention in different contexts and settings is needed. Also, due to the importance of the rurality factor, it would be appropriate to analyze those rural areas or developing countries where socio-economic and socio-demographic variables may have a negative impact on the implementation of certain methodologies. A comprehensive approach to Early Intervention has also been addressed, but future research could also focus on studying both types of intervention in populations with specific diagnoses. The ultimate aim is to improve services by providing the necessary training to practitioners and support to families and children, from a preventive and intervention point of view.

Finally, the development of new assessment tools validated within the context of tele-intervention could be fertile ground for the development and implementation of regulatory policies that benefit the early childhood intervention community.

## Figures and Tables

**Figure 1 healthcare-12-00112-f001:**
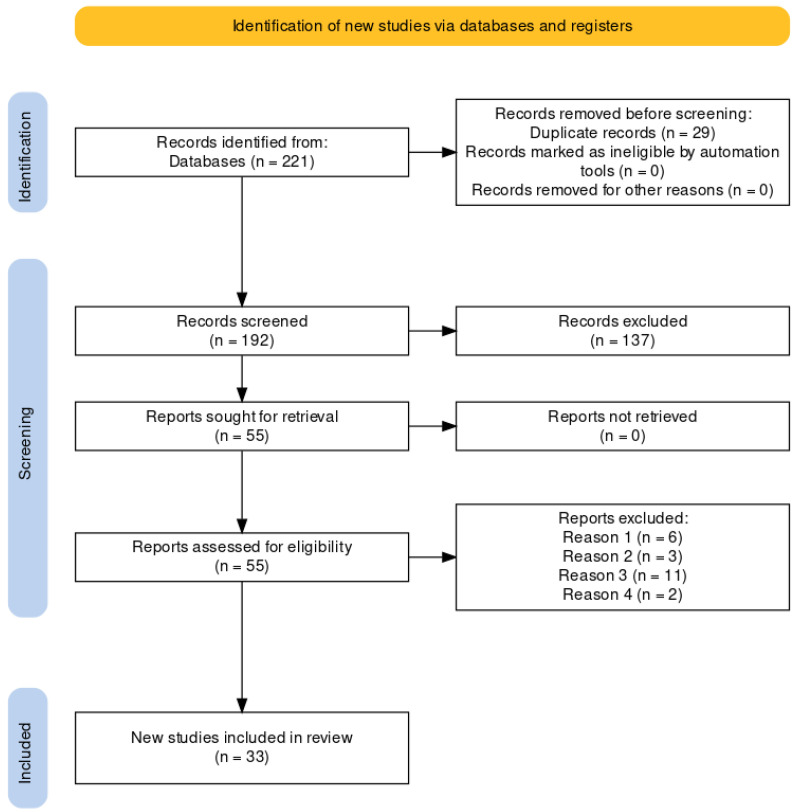
Flowchart for the identification of new studies through databases and registers according to PRISMA. Reason 1: selection criterion 6; Reason 2: selection criterion 5; Reason 3: selection criterion 8; Reason 4: selection criterion 4.

**Figure 2 healthcare-12-00112-f002:**
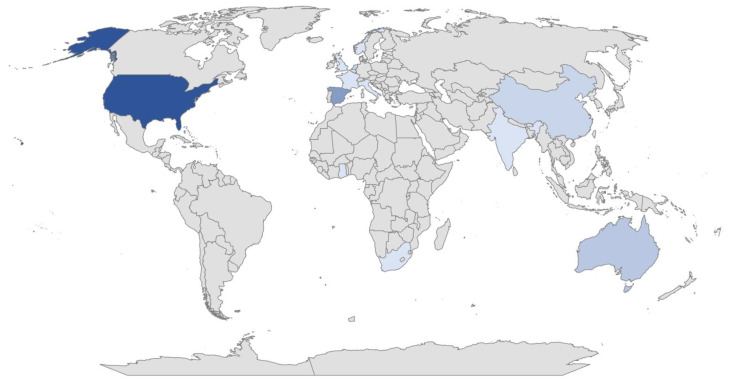
Geographical territory where the research selected for this study has been carried out worldwide. Colors of higher intensity represent higher density values.

**Figure 3 healthcare-12-00112-f003:**
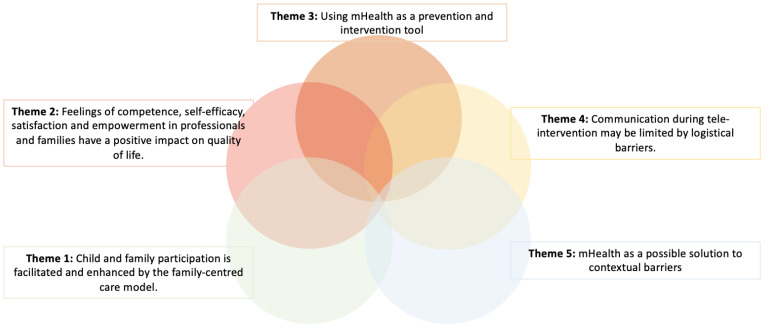
Representation of the five emerging themes from the studies included in this systematic review.

**Figure 4 healthcare-12-00112-f004:**
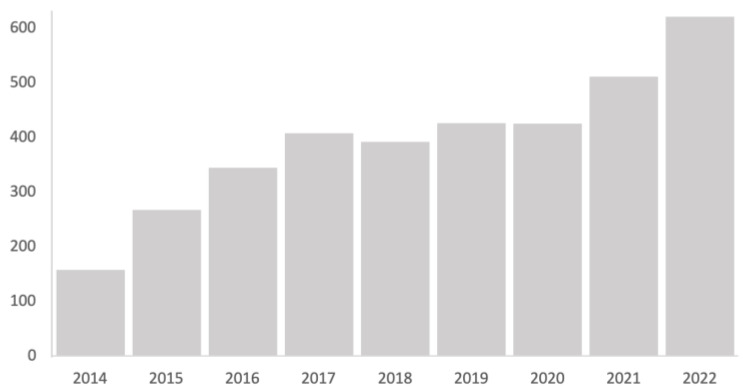
Trends in the number of publications in Web of Science that include the term “mHealth” in the title in the period 2014–2022. Source: Web of Science (WOS).

**Table 1 healthcare-12-00112-t001:** Extraction record and description of the studies reviewed.

Title	Authors and Year	Country	Type of Design	Sample	Variables Studied	Conclusions
Even though a lot of kids have it, not a lot of people have knowledge of it: A qualitative study exploring the perspectives of parents of children with cerebral/cortical visual impairment.	Oliver et al., 2023 [41]	USA	Mixed: qualitative and quantitative ex post facto descriptive.	Parents with children diagnosed with cortical visual impairment (CVI)	Awareness of CVI and its impact on children and family; parental experience; child factors and functional implications; supports that enhance vision development.	Need for education on the importance of early diagnosis, the clinical features of CVI and adaptations to optimize the child’s progress. This is related to more positive experiences for parents with less of a burden and frustration, and a feeling that they could actively collaborate in the process.
Social validity of telepractice in early intervention: Effectiveness of family-centered practices.	Martínez-Rico et al., 2023 [42]	Spain	Ex post facto cross-sectional quantitative.	659 Spanish families from 35 EI centers.	Social validity (usability, effectiveness (competence and confidence), intervention with the caregiver, feasibility, usefulness and future interventions).	Active family participation and attention to the needs and priorities that they require are relevant factors for social validity in Spanish families in EI through tele-intervention. Other key components for this sphere of intervention that have a significant influence on the perception of families are also highlighted; collaborating to find joint solutions and promoting active participation during the sessions.
Diversity of Child and Family Characteristics of Children with Hearing Loss in Family-Centered Early Intervention in The Netherlands.	Van der Zee and Dirks, 2022 [43]	The Netherlands	Quantitative descriptive cross-sectional ex post facto.	Dutch children born in 2014–2016 with bilateral hearing loss who are in the FCEI program.	Socio-demographic factors, language, intervention-related factors and family involvement.	Despite having a common diagnosis, the characteristics of the children and families were very heterogeneous. Need to promote an intervention taking into account the beliefs and needs of the whole family, supporting and guiding them. The involvement of the family and the appearance of additional disabilities were predictive factors of the children’s language abilities.
Family-centered practices in early intervention: family confidence, competence, and quality of life.	Subinas-Medina et al., 2022 [44]	Spain	Quantitative cross-sectional ex post facto, correlational and descriptive.	43 Spanish families of children attending the EI service.	Family quality of life; competence and confidence of families, number of practitioners, parental confidence and competence.	Families receiving EI services in Spain have a fairly good perception of FQoL, which may be related to the support of a single practitioner. The higher the confidence in parenting and parental competence, the higher the family quality of life; the confidence of primary caregivers in helping the family predicts the confidence and competence of parents in supporting the child in daily routines.
Assessing the Satisfaction and Acceptability of an Online Parent Coaching Intervention: A Mixed-Methods Approach.	Qu et al., 2022 [45]	China	Mixed: qualitative and randomized clinical trial.	32 caregivers with children diagnosed with ASD aged 2–5 years old.	Acceptability, timeliness, feasibility, project-level suggestions and service-level considerations.	Positive perceptions regarding the variables of satisfaction, acceptability, suitability, feasibility and recommendation of the program.
Communicating With Intention: Therapist and Parent Perspectives on Family-Centered Care in Early Intervention.	Popova, O’Brien and Taylor, 2022 [21]	USA	Ex post facto cross-sectional quantitative.	101 therapists (developmental n = 29; occupational n = 32; physical n = 17; speech n = 28) and 19 parents involved in the EI program.	Self-efficacy (professionals and parents); family-centered intervention process; therapeutic communication; suboptimal interactions; sociodemographic variables.	The family-centered model is a benchmark in pediatrics that ensures parental engagement and is associated with positive therapeutic outcomes. Therapists’ capacity for effective application of this model is limited. The Intentional Relationship Model recognizes that the interpersonal dynamic between therapist–parents–child has the power to enable or inhibit parent and child engagement in therapy. The distinction between different ways therapists communicate can implement an intervention based entirely on the family-centered model.
Impact of a family-centred early intervention programme in South India on children with developmental delays.	Muthukaruppan et al., 2022 [46]	India	Quantitative: Open prospective longitudinal cohort design.	308 primary caregivers of children aged 0–6 years with developmental delay who were receiving some form of care within the study program.	Family empowerment and level of caregiver burden	The study demonstrates that early family-centered intervention, supported by digital technology to provide training and education to caregivers, has positive effects on empowerment and burden level. Change evident in all caregivers regardless of sex, age or place of therapy.
Relationships between family-centered practices and parent involvement in early childhood intervention.	Mas et al., 2022 [47]	Spain	Quantitative: Quasi-experimental design	278 parents of children aged 0–6.	Socio-demographic characteristics; family-centered participatory practices; family-centered relational practices and level of participation.	Meaningful and active participation in EI has greater benefits than merely relational practices.
Family-Centered Early Intervention (FCEI) Involving Fathers and Mothers of Children Who Are Deaf or Hard of Hearing: Parental Involvement and Self-Efficacy.	Dirks & Szarkowski, 2022 [48]	The Netherlands	Quantitative cross-sectional post facto.	24 Dutch couples (mother-father) with at least one child with moderate-severe hearing loss aged 0–4 years.	Self-efficacy in parenting; experience with children with hearing loss; Perceived support from family-centered EI services and Frequency of participation in family-centered EI. Family.	Although both fathers and mothers reported high levels of self-efficacy, mothers reported higher levels than fathers in some of the domains. There was a tendency for mothers to be more involved in the interventions. All this points to possible differences in terms of needs, and therefore providers should address these needs differently.
An mHealth Intervention to Reduce the Packing of Discretionary Foods in Children’s Lunch Boxes in Early Childhood Education and Care Services: Cluster Randomized Controlled Trial.	Pearson et al., 2022 [49]	Australia	Cluster randomized controlled trial.	355 parent–child dyads for 3–6 year-olds from 17 education and EI services.	Packaging and consumption of packed lunch food; characteristics of parents and service; process evaluation.	Despite the intervention not achieving the main objective, the following data were obtained: the use of apps was rated as an adequate modality to deliver information, lack of impact associated with the implementation of the intervention, low acceptance and use of the apk by parents.
Examining early intervention referral patterns in neonatal intensive care unit follow-up clinics using telemedicine during COVID-19.	Miller et al., 2022 [50]	USA	Ex post facto cross-sectional quantitative.	658 NICU follow-up visits (384 face-to-face and 274 tele-intervention).	Medical referral pattern; school referral pattern; distance travelled home-clinical centre; level of satisfaction with the service	All babies received the necessary interventions, reducing the cost and distance travelled in remote areas. Likewise, referral rates were significantly higher for medical services and the same likelihood of referral for school services. It is concluded that telemedicine saves time, money and is as effective or better at identifying the need for additional referral.
Neonatal-Assisted Telerehabilitation (T.A.T.A. Web App) for Hearing-Impaired Children: A Family-Centered Care Model for Early Intervention in Congenital Hearing Loss.	Landolfi et al., 2022 [51]	Italy	Quantitative longitudinal.	15 children with deafness (240–300 days).	Socio-demographic characteristics; hearing skills; Family involvement in the intervention program.	The TATA app provides proactive management of DHH children through parental involvement. It provides a general and specific developmental profile of emerging skills and at-risk situations. Alongside this, other research has shown that this model facilitates family inclusion; essential for improving children’s language outcomes and enabling children’s continued education in their routine settings.
mHealth-Supported Hearing Health Training for Early Childhood Development Practitioners: An Intervention Study.	Du Plessis et al., 2022 [52]	South Africa	Experimental design. Randomized pre–post test clinical design.	1012 practitioners aged 17–31.	Legibility of the information provided; training; knowledge post 6 months.	It is further concluded that a multimedia mHealth auditory training program is a scalable, low-cost intervention to provide professionals with the necessary knowledge to identify and refer children at risk and to support children with difficulties in the school environment.
Early Detection of Neurodevelopmental Disorders of Toddlers and Postnatal Depression by Mobile Health App: Observational Cross-sectional Study.	Denis et al., 2022 [53]	France	Quantitative cross-sectional.	4000 users of the Malo apk.	Age and sex of children; neurodevelopmental skills; language, socialization, hearing, vision and motor skills; risk of postpartum depression; relevance of physician alerts and level of satisfaction.	This multi-domain apk dedicated to the early detection of NDD and PND suggests that a multi-domain family mHealth app is suitable and effective for regular use in monitoring the mother–child dyad. It should be noted that the main finding of this study is that 0.9% of small children were identified as potential ASD children at 11 months; this is very close to the ASD rate in the general population (0.6%) with a mean age of detection of 4–6.8 years.
A Qualitative Study Exploring Parental Perceptions of Telehealth in Early Intervention.	Cheung et al., 2022 [54]	USA	Qualitative	15 parents of children who had received EI via telehealth (1–3a).	Parent–child socio-demographic data; Access to telehealth EI services; Practitioner–family partnerships; Training; Initial support for facilitating EI through telehealth.	Four main findings: advantages and disadvantages of accessing EI services through tele-intervention; high-quality family–practitioner partnerships preserved during sessions; need to overcome logistical barriers to accessing this modality of services; practitioners need to strengthen their knowledge and skills on how to collaborate and empower caregivers during sessions rather than carry out direct therapy on the child.
Involving Caregivers of Autistic Toddlers in Early Intervention: Common Practice or Exception to the Norm?	Lee et al., 2022 [55]	USA	Quantitative, cross-sectional ex post facto and descriptive.	Families with children under 36 months with a diagnosis of ASD.	Socio-demographic characteristics; behaviours observed during the therapy session.	The creation of strong bonds between the practitioner who treats language disorders and the family.
Brief Report: Perceptions of Family-Centered Care Across Service Delivery Systems and Types of Caregiver Concerns About Their Toddlers’ Development.	Dick et al., 2022 [56]	USA	Part of a longitudinal study.	Family members of children with ASD 16–33 months (n = 37); family members of children with other neurodevelopmental disorders (n = 22).	Socio-demographic data; measurement of the care process distinguishing between primary and early intervention services.	Caregivers perceive higher levels of family-centered care from early intervention professionals than from primary care professionals. The importance of both teams working continuously during the detection, diagnosis and intervention processes is underlined.
mHealth Intervention for Motor Skills: A Randomized Controlled Trial.	Staiano et al., 2022 [57]	USA	Randomized controlled study.	n = 72 children (3–5 years); 35 motor skills app and 37 free play app.	Motor skills; degree of acceptability and adherence.	The use of the motor skills apk led to an increase in the motor skills percentile. Likewise, there were also improvements in motor skills that were not covered by the apk, indicating a transfer of learning to more global aspects. High levels of adherence and acceptability were recorded in both applications.
Early Intervention Services During the COVID-19 Pandemic in Spain: Toward a Model of Family-Centered Practices.	Vilaseca et al., 2021 [58]	Spain	Ex post facto cross-sectional quantitative.	Sub-sample 1.—81 families of children cared for in EI (0–6); Sub-sample 2.—213 professionals working in EI.	Perception of families on the change in intervention methodology after the pandemic; Perception of professionals on the changes in intervention methodology with families and children after the pandemic; Socio-demographic variables.	The change in methodology did not present significant changes in terms of incorporating the family-centered model. Professionals considered that the intervention followed the trends of this model, but the results of the families were inconclusive, reflecting the difficulty of application with respect to socio-demographic variables.
Service Quality in Early Intervention Centres: An Analysis of Its Influence on Satisfaction and Family Quality of Life.	Jemes-Campaña et al., 2021 [59]	Spain	Ex post facto cross-sectional quantitative.	1551 parents of children attending one of the 24 EI centers in Andalusia participating in the study with developmental problems or at risk of developmental problems.	Degree of satisfaction with the service; Perceived quality of the service; Family quality of life.	Perceived quality and satisfaction with EI centers are tools for achieving family quality of life. There are relationships between these aspects, which, together with the degree of support received by families from professionals, influence quality of life.
Family-centered early intervention: Comparing practitioners’ actual and desired practices.	García-Ventura et al., 2021 [60]	Spain	Ex post facto cross-sectional quantitative.	119 Spanish EI practitioners whose projects based on the family-centered model were in the early stages.	Practitioner characteristics; Current and desired practices.	There is a desire to move towards the family-centered model in current practices on the part of practitioners, but there are also barriers to this that are not dependent on the practitioner. Likewise, no correlation is found between years of experience or level of university studies with FOCAS results.
Does Parental Education Level Matter? Dynamic Effect of Parents on Family-Centered Early Intervention for Children with Hearing Loss.	Chen & Liu, 2021 [61]	Taiwan	Mixed: Path Analysis.	113 children 3–6 years old with permanent bilateral hearing loss attending auditory–verbal therapy in an EI centre in Taiwan.	Language skills (language comprehension and speech); time in hearing therapy; age at which hearing aids are used and parents’ educational levels.	The role of parents has clinical implications for language comprehension in children with hearing loss, meaning that cooperation with both parents is key. Similarly, social disadvantages between caregiver and child could be reduced through early intervention.
Family Perspectives toward Using Telehealth in Early Intervention.	Yang et al., 2021 [62]	USA	Qualitative.	37 parents with at least one child aged 0–9 who has received EI assistance.	Socio-demographic data; Family perceptions on the use of telehealth; Family perceptions on the advantages of telehealth; Family perceptions on the disadvantages of telehealth; Family perceptions on logistical elements needed to implement telehealth-based care.	Participants were reluctant to use telehealth in EI. The possible explanation could be misconceptions about the aims and purposes of EI as well as logistical barriers to accessing services and materials.
Practitioners’ Self-Assessment of Family-Centered Practice in Telepractice Versus In-Person Early Intervention.	McCarthy et al., 2021 [20]	Australia	Ex post facto cross-sectional quantitative.	52 early intervention professionals and 239 children under 8 years old.	Socio-demographic data; rating of care processes for service providers (MPOC-SP).	Practitioners who worked via telepractice reported using family-centered practices to a similar degree as those who intervened face to face. Analyses of the MPOC-SP scale indicated that there were no significant relationships between practitioners’ assessment of their use of family-centered practices and mode of intervention; results were consistent even with other more specific variables such as type of practitioner, experience, etc.
Early intervention service delivery via telehealth during COVID-19: a research–practice partnership.	Kronberg et al., 2021 [63]	USA	Quasi-experimental design with pre–post measures.	17 families enrolled in the state EI program aged 6–34 months.	Socio-demographic data; progress towards achieving the family’s goals; Goal performance and degree of satisfaction.	The findings suggest that a 9-week coaching intervention provided through telehealth by community-based specialist EI practitioners can be effective in promoting parents’ goals and satisfaction motivated by their children’s achievements.
Impact of Family-Centered Early Intervention in Infants with Autism Spectrum Disorder: A Single-Subject Design.	Park et al., 2020 [64]	Republic of Korea	Case study.	3 children aged between 2 and 3 years with suspected ASD.	Interventions based on the family-centered model of care (environmental modifications, video recording, task training and feedback, individualized information on the child and task completion rate); Frequency of social interactions during the interventions; changes in social interaction skills; changes in ASD risk.	After implementing the family-centered early intervention program, all participants improve significantly during and after the intervention in the three modes of social interaction skills (appearance, gestures and speech). The ASD risk score decreased significantly. Similarly, parents’ performance improved and, with it, their internal motivation.
Early Childhood Intervention Program Quality: Examining Family-Centered Practice, Parental Self-Efficacy and Child and Family Outcomes.	Hughes-Scholes & Gavidia-Payne, 2019 [65]	Australia	Clinical trial with pre–post measures.	66 families with children with developmental problems (average age 46 months).	Socio-demographic characteristics; family outcomes; parental self-efficacy and perceptions of practices before and after the intervention.	There is an improvement in both child and parents’ skills following participation in the family-centered EI program, although no direct links can be made between this improvement and the model itself.
Proactive Assessment of Obesity Risk during Infancy (ProAsk): a qualitative study of parents’ and professionals’ perspectives on an mHealth intervention.	Rose et al., 2019 [66]	United Kingdom	Qualitative.	66 families of children aged 6–8 weeks and 22 health visitors.	Participation and empowerment with digital technology; Unfamiliar technology presents challenges and opportunities; Confidence in risk scoring; Resistance to targeting.	The intervention based on the mHealth model actively involved parents, allowing them to take ownership of the process of finding strategies to reduce the risk of childhood overweight. Cognitive and motivational biases were detected that prevented effective communication on the central theme, causing barriers in the intervention of those infants most at risk.
Interregional Newborn Hearing Screening via Telehealth in Ghana.	Ameyaw, Ribera & Anim-Sampong, 2019 [67]	Ghana	Non-randomized cross-sectional quantitative	50 nursing infants aged 2–90 days (convenience sample; 30 males and 20 females).	Traditional screening procedure, virtual screening procedure, duration of both procedures.	The study shows the possible feasibility of establishing an interregional network of newborn hearing screening services in Ghana using telehealth, demonstrating high efficiency rates when comparing the use of these services with the mobility of these families to receive similar services.
Early food for future health: a randomized controlled trial evaluating the effect of an eHealth intervention aiming to promote healthy food habits from early childhood.	Helle et al., 2017 [68]	Norway	Randomized controlled study.	718 Norwegian parents with a full-term child of 3–5 months of age with a birth weight ≥ 2500 g.	In children: anthropometric measures; food intakes; food variation; child’s eating behaviour; child temperament, food preferences; child behavior. In parents: socio-demographic characteristics; anthropometric measures; food intake; food variation; food neophobia; feeding style and feeding practices; feeding self-efficacy; parenting style; personality traits and mental health.	The Early Food for Future Health eHealth intervention guides parents of children aged 6–12 months through the different developmental stages related to feeding. Its use can increase awareness and understanding of the importance of preventing childhood overweight and obesity in terms of: design and effectiveness of internet-based interventions and the relationship between parenting, feeding behavior in parents and children.
Text messaging data collection for monitoring an infant feeding intervention program in rural China: feasibility study.	Li et al., 2013 [69]	China	Quantitative with pre–post measures non-randomized.	258 participants (n = 99; text messaging respondents vs. n = 177 face-to-face respondents).	Response rate; Data agreement; Costs; Acceptability of text messages and face-to-face surveys; Reasons for not responding to text messages; Reasons for disagreement between survey methods; Adequate moment to send text messages.	The feasibility of using text messaging as a method of data collection for monitoring health programs in rural China was studied. Although the text message survey was acceptable and there was a reduction in costs, it had a lower response rate. Future research is needed to evaluate the effectiveness of strategies to increase the response rate, especially in terms of longitudinal data collection.
Measuring Costs and Outcomes of Tele-Intervention When Serving Families of Children who are Deaf/Hard-of-Hearing.	Blaiser et al., 2013 [70]	USA	Randomized controlled trial.	27 families enrolled in the Utah Schools for the Deaf and the Blind (USDB) Parent Infant Program (PIP).	Receptive and expressive language skills; costs; quality of the home visit; perceptions of carers and providers.	One of the reasons why some families did not want to participate is that the use of tele-intervention implied an additional effort associated with learning an NT that was not desirable or feasible at that time. This aspect to which participants were subjected during the first months of the study influenced the satisfaction scores. Another aspect to keep in mind is that satisfaction improved dramatically when connection problems were solved; for the implementation of these services, access to sufficient bandwidth is necessary, which translates into upload and download speed.
Overview of States’ Use of Telehealth for the Delivery of Early Intervention (IDEA Part C) Services.	Cason et al., 2012 [71]	USA	Quantitative ex post facto.	Representatives from 26 states and one IDEA Part C jurisdiction.	Early telehealth intervention providers; telehealth reimbursement within early intervention; barriers to telehealth implementation.	Many states are incorporating telehealth into their care programs under the Early Intervention for Individuals with Disabilities Education Act to improve service quality. Practitioners under the Act are already using telehealth services to provide habilitation services and specialized consultations (IDEA). Policy development, education, research, the use of secure applications and the promotion of strategies are important in order to qualify telehealth as a service model within IDEA Part C programs.

Note. EI = early intervention; App = mobile app; FQoL = family quality of life; CVI = cortical visual impairment; DHH = Deaf or Hard of Hearing; FCEI = family-centered early intervention; FOCAS = family orientation of community and agency services scale; MPOC-SP = Measure of Processes of Care for Service Providers; NDD = neurodevelopmental disorders; NT = new technologies; PND = postnatal depression; ASD = autism spectrum disorder; NICU = neonatal intensive care unit.

**Table 2 healthcare-12-00112-t002:** Level of evidence and grade of recommendation according to SIGN.

Authors and Year	Level of Evidence	Grade of Recommendation
Oliver et al. (2023) [41]	2++	B
Martínez-Rico et al. (2023) [42]	2++	B
Van der Zee & Dirks (2022) [43]	2++	B
Subinas-Medina et al. (2022) [44]	2++	B
Qu et al. (2022) [45]	1++	A
Popova et al. (2022) [21]	2++	B
Muthukaruppan et al. (2022) [46]	2+	C
Mas et al. (2022) [47]	2++	B
Dirks & Szarkowski (2022) [48]	2+	C
Pearson et al. (2022) [49]	1++	A
Miller et al. (2022) [50]	2++	B
Landolfi et al. (2022) [51]	2+	C
Du Plessis et al. (2022) [52]	1++	A
Denis et al. (2022) [53]	2++	B
Cheung et al. (2022) [54]	2−	C
Lee et al. (2022) [55]	2++	B
Dick et al. (2022) [56]	2++	B
Staiano et al. (2022) [57]	1++	A
Vilaseca et al. (2021) [58]	2++	B
Jemes-Campaña et al. (2021) [59]	2++	B
García-Ventura et al. (2021) [60]	2++	B
Chen & Liu (2021) [61]	1+	A
Yang et al. (2021) [62]	2+	C
McCarthy et al. (2021) [20]	2++	B
Kronberg et al. (2021) [63]	2+	C
Park et al. (2020) [64]	2-	C
Hughes-Scholes & Gavidia-Payne (2019) [65]	2+	C
Rose et al. (2019) [66]	2+	C
Ameya et al. (2019) [67]	1++	A
Helle et al. (2017) [68]	1++	A
Li et al. (2013) [69]	2+	C
Blaiser et al. (2013) [70]	1++	A
Cason et al. (2012) [71]	2++	B

Note. (1++) High-quality meta-analyses, systematic reviews of randomized control trial (RCT), or RCTs with a very low risk of bias; (1+) Well-conducted meta-analyses, systematic reviews, or RCTs with a low risk of bias; (2++) High-quality systematic reviews of case control or cohort or studies or High-quality case control or cohort studies with a very low risk of confounding or bias and a high probability that the relationship is causal; (2+) Well-conducted case control or cohort studies with a low risk of confounding or bias and a moderate probability that the relationship is causal; (2−) Case control or cohort studies with a high risk of confounding or bias and a significant risk that the relationship is not causal. The letters A, B and C are directly correlated with levels of evidence.

## Data Availability

Data are contained within the article. No additional data are available.

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
