# Peer review of "Evidence-Based Implementation of the Family-Centered Model and the Use of Tele-Intervention in Early Childhood Services: A Systematic Review"

_healthcare, 2024, doi:10.3390/healthcare12010112_

Round 1

Reviewer 1 Report

Comments and Suggestions for Authors

Good work. The authors have done a good job of conducting a systematic review. The use of the English language is appropriate. 

There are some issues that I would like to address:

1) The title gives the impression that family-centred model and the teleintervention models are the same thing. But, they are different. so, the title should reflect this independence of the two models.

2) section 1.1. This Study is not clearly written. Words with similar meanings were used in this section: aim, goal and objective. I feel that the authors should re-write the section to make it more understandable. The research objective should be clarified.

3) There is a major confusion in the writing of the teleintervention model. I finally learned that MHealth and the teleintervention model are the almost the same concepts. I think the authors should use either of these terms throughout the paper. 

4) Regarding Table 2, under the table, you need to have a key for the level of evidence and grade of recommendation. Please use a short note to describe the level of evidence and the grade of recommendation. Yes, we have a description in earlier pages but we need to have another short description; for example we should know the answer of "what is 2++?".

5) Lines 241-246: It is not clear whether these abbrevations are belonged to this spot.

6) Lines 160-164: These statements are unclear. Please re-write. 

7) A major question: Why did you review the research on two different models? Why did not you prefer one or the other? Just the family-centered or the teleintervention model could be reviewed. I do not suggest that there should be just a single method; but, your rationale should be clear and clarified in the paper. 

8) There are overlaps between lines 165-172 and lines 173-182. There are repetitions where you describe the inclusion and exclusion criteria. You may want to combine two criteria list to have a single paragraph. 

9) Lines 187-189: Expand on the content analysis. How did you conduct the content analysis?

10) Lines 194-196: How did you reach the 55 publications out of the 255 research works? It should be addressed in-depth. 

11) Lines 228-234: How did you conduct the synthesizing process?

12) Figure 4: You need to reverse the order of the years on the x-axis. It should begin from 2014 and to 2022. 

Thanks. 

Author Response

Dear reviewer,

Thank you very much for your contributions. In the attached PDF document, you will find the response to each of the questions raised. The corresponding changes are reflected in the new version of the submitted manuscript.

Best regards.

Reviewer 2 Report

Comments and Suggestions for Authors

Comments on the Quality of English Language

Minor editing of English language would be good for the overall quality of the article.

Author Response

(The authors gave the same response as above.)

Reviewer 3 Report

Comments and Suggestions for Authors

1. It would be helpful if the authors provide more information regarding how the levels of evidence and grade of recommendation were decided for the studies included in this systematic review.

2. It would be helpful if the authors provide and synthesize the effect sizes of the quantitative studies. The effect sizes would support the authors statements regarding effectiveness of the family-centred model and teleintervention in early childhood services.

3. Are effect sizes of the interventions using the family-centred model larger than those using the child-centred model?

4. The authors need to examine potential publication bias.

5. The authors should consider examine moderators of the effects of the interventions. For example, are teleinterventions more effective for samples with low SES?

Comments on the Quality of English Language

The quality of English language is good with only minor errors that require proofreading. 

Author Response

(The authors gave the same response as above.)

Round 2

Reviewer 2 Report

Comments and Suggestions for Authors

Thank you for all the revisions.  The manuscript is much better and appropriate for publish.